**Data Availability Statement:** There is legal restriction on sharing the de-identified dataset based on the Master Services Agreement for

# Using medical claims database to develop a population disease progression model for leuprorelin-treated subjects with hormone-sensitive prostate cancer

Yixuan Zou[1,2], Fei Tang🄳[1], Jeffery C. Talbert[3], Chee M. Ng[1,4]*

**1** Department of Pharmaceutical Sciences, College of Pharmacy, University of Kentucky, Lexington, KY, United States of America, **2** Department of Statistics, University of Kentucky, Lexington, KY, United States of America, **3** Department of Pharmacy Practice and Science, College of Pharmacy, University of Kentucky, Lexington, KY, United States of America, **4** NewGround Pharmaceutical Consulting LLC, Foster City, CA, United States of America

* cheemng@gmail.com

## Abstract

Androgen deprivation therapy (ADT) is a widely used treatment for patients with hormone-sensitive prostate cancer (PCa). However, duration of treatment response varies, and most patients eventually experience disease progression despite treatment. Leuprorelin is a luteinizing hormone-releasing hormone (LHRH) agonist, a commonly used form of ADT. Prostate-specific antigen (PSA) is a biomarker for monitoring disease progression and predicting treatment response and survival in PCa. However, time-dependent profile of tumor regression and growth in patients with hormone-sensitive PCa on ADT has never been fully characterized. In this analysis, nationwide medical claims database provided by Humana from 2007 to 2011 was used to construct a population-based disease progression model for patients with hormone-sensitive PCa on leuprorelin. Data were analyzed by nonlinear mixed effects modeling utilizing Monte Carlo Parametric Expectation Maximization (MCPEM) method in NONMEM. Covariate selection was performed using a modified Wald's approximation method with backward elimination (WAM-BE) proposed by our group. 1113 PSA observations from 264 subjects with malignant PCa were used for model development. PSA kinetics were well described by the final covariate model. Model parameters were well estimated, but large between-patient variability was observed. Hemoglobin significantly affected proportion of drug-resistant cells in the original tumor, while baseline PSA and anti-androgen use significantly affected treatment effect on drug-sensitive PCa cells ($Ds$). Population estimate of $Ds$ was 3.78 x 10$^{-2}$ day$^{-1}$. Population estimates of growth rates for drug-sensitive ($Gs$) and drug-resistant PCa cells ($G_R$) were 1.96 x 10$^{-3}$ and 6.54 x 10$^{-4}$ day$^{-1}$, corresponding to a PSA doubling time of 354 and 1060 days, respectively. Proportion of the original PCa cells inherently resistant to treatment was estimated to be 1.94%. Application of population-based disease progression model to clinical data allowed characterization of tumor resistant patterns and growth/regression rates that enhances our understanding of how PCa responds to ADT.

Health Services Collaboration between the University of Kentucky and Comprehensive Health Insights, Inc. In this agreement, the company (Comprehensive Health Insights) provided a "De-Identified Historical Dataset" to the university (University of Kentucky), and the university could only use this dataset for educational and training purposes. An independent researcher who would like to request the data would need to contact Humana (Vinit Nair (vnair1@humana.com)). The data dictionary for the Humana database is included in the Supporting Information. A non-author institutional contact is Tammy Harper (Tamela.harper@uky.edu), the project manager for the University of Kentucky Center for Clinical and Translational Science (CCTS) Enterprise Data Trust.

**Funding:** CN is supported in the form of a salary from NewGround Pharmaceutical Consulting LLC. The funders had no role in study design, data collection and analysis, decision to publish, or preparation of the manuscript. The specific roles of these authors are articulated in the 'author contributions' section.

**Competing interests:** CN is an employee of NewGround Pharmaceutical Consulting LLC. There are no patents, products in development or marketed products to declare. This does not alter our adherence to PLOS ONE policies on sharing data and materials.

## Introduction

Prostate cancer (PCa) is one of the most common cancers among men in the United States, accounting for 44% of all cancer cases along with lung and bronchus, and colorectal cancers [1]. In 2016, there were approximately 180,890 new cases of PCa and 26,120 deaths due to PCa in the United States [1]. Clinically localized PCa is most commonly managed by observation, radical prostatectomy, and radiotherapy (with or without androgen deprivation therapy, or ADT) [2]. However, many patients diagnosed with localized disease ultimately undergo bio-chemical progression as demonstrated by increasing levels in prostate-specific antigen (PSA), who may then be treated by ADT. ADT acts by depleting gonadal testosterone [3], and it can be achieved by medical castration through the use of luteinizing hormone-releasing hormone (LHRH) agonists or antagonists, or surgical castration (bilateral orchiectomy), both of which are considered equally effective. On the other hand, for patients who present with metastatic PCa, ADT is considered the gold standard of initial therapy that has been shown to reduce tumor-related events [4, 5]. ADT is a widely used systemic treatment for patients with hormone-sensitive PCa, including those who previously receive local therapy as the primary treatment modality and those who present with *de novo* metastatic disease. However, the majority of patients eventually progress to a castration-resistant state, and it was estimated that the response to ADT typically lasts 14 to 20 months in metastatic PCa [5, 6], and the duration of response to ADT is variable among patients [7]. A few studies have examined clinical factors that predict the time to castration resistance for patients on ADT [8–12], but the detailed time-dependent profile of tumor regression and/or growth in patients with hormone-sensitive PCa on medical ADT has never been fully characterized.

Quantitative population-based disease progression modeling uses mathematical functions and statistical models to describe quantitatively the time course of disease progression in individuals and the entire patient population with or without drug treatment [13, 14]. This modeling approach can trace disease progression over time, and quantify the effects of drug treatment and disease- and patient-specific factors on disease progression to optimize clinical trial design and guide personalized treatment strategies [13]. For example, quantitative disease progression modeling has been applied to clinical data to allow estimation of tumor growth and regression rates in metastatic castration-resistant PCa [14]. Most of the published quantitative population-based disease progression models are developed using data from randomized controlled studies [13–15]. While randomized controlled trials are the golden standard for evaluating new drug treatment and the primary source of clinical research information, they are costly, labor-intensive and time-consuming [16]. In addition, strict inclusion and exclusion criteria are typically used to select subjects in randomized clinical trials within a pre-defined study period, and thus the study results may have limited utility in answering clinical questions in a real-life population [16, 17]. These limitations of randomized controlled trials have led to an increased reliance on using medical claims data in designing healthcare policies related to real-life clinical practice [16, 17]. Medical claims data are a rich and inexpensive source of scientific information to study how a disease responds to medical interventions of interest, and can provide an opportunity to potentially follow a large number of patients for extended periods of time without suffering from high attrition rates [16–18]. To our knowledge, medical claims data have never been used to develop quantitative population-based disease progression models, in which effects of drug treatment and patient-specific factors on disease progression are evaluated and quantified.

The primary goal of this study was to use a medical claims database to construct a quantitative population-based disease progression model for patients with hormone-sensitive PCa on ADT with LHRH agonists. Leuprorelin, the most commonly prescribed LHRH agonist in the

Medicare population [19], was selected as the treatment agent of interest in this analysis. Prostate specific antigen (PSA) is an androgen-regulated serine protease produced almost exclusively by cells in the prostate gland [20, 21]. PSA is commonly assessed for PCa screening, and it remains an important marker for monitoring clinical response to ADT treatment [21–24]. Therefore, PSA was used as a molecular marker of tumor burden to measure disease progression of PCa in this study.

## Materials and methods

### Data source

This study was conducted in commercially insured patients using health claims data provided by Humana covering the period from January 1, 2007 to December 31, 2011. This nationwide database captures anonymized longitudinal, individual-level data on patient demographics, healthcare utilization, inpatient and outpatient diagnostic and procedural codes, laboratory test results, and pharmacy dispensing records for more than 8.1 million commercially insured people in the United States. This study was approved by the Institutional Review Board at the University of Kentucky, and the requirement to document informed consent was waived. All data used in this study were anonymized and deidentified. Data retrieved from SQL queries were pre-processed in R software (version 3.4.3, The R Foundation for Statistical Computing, Austria) for dataset merging, wrangling and formatting (packages used: 'readr', 'dplyr', 'stringr', 'ggplot2', 'data.table').

### Study subjects

Subjects were selected based on the following criteria: 1) diagnosis with malignant PCa (ICD-9-CM code 185 or ICD-10-CM code C61), 2) continuous or intermittent use of leuprorelin as the only agent among LHRH agonists, and 3) availability of a PSA level before initiation of leuprorelin (i.e, a baseline PSA level) and at least one level during treatment.

The following patients were excluded due to inability of our disease progression model to describe certain unusual PSA profiles: 1) PSA levels decreased over time prior to initiation of leuprorelin treatment, the baseline PSA level was undetectable, or there were multiple measurements reported on the same day; 2) subjects had undetectable PSA levels throughout leuprorelin treatment. Patients were also excluded if 3) they had incomplete demographic data (age, region and race). Lastly, patients were excluded if 4) they had extremely low hemoglobin levels ($< 6$ g/dL), as acute illness was likely involved (S1 Fig).

For all selected subjects, PSA levels starting from baseline to the last available measurement or before the initiation of continuous antiandrogen therapy, surgery, radiotherapy or chemotherapy (whichever occurred first) were included for model development. For each subject, race, age, region and the use of antiandrogens (bicalutamide, enzalutamide, flutamide, nilutamide) within 30 days of leuprorelin initiation (presumably for preventing flare reactions associated with leuprorelin treatment [6]) were extracted for model development. In addition, laboratory measurements including those that have been reported to affect clinical outcomes in castration-resistant PCa were extracted for each patient. These laboratory measurements included aspartate transaminase (AST), alanine transaminase (ALT), serum creatinine (SCR), alkaline phosphatase (ALP), albumin (ALB) and hemoglobin (HGB) [8, 9, 25–33].

### Structural model

All modeling steps were conducted in NONMEM (ver. 7.3, ICON Development Solutions, USA). Two mathematical models with different resistance development patterns were

developed to describe the observed PSA kinetics. In the first model (Model I), drug resistance was developed from PCa tumor cells that were initially sensitive to LHRH treatment and caused by adaptive responses, such as target gene mutations, altered expression of therapeutic targets, and stimulation of compensatory signaling pathways [34]. The following equation modified from Wilkerson et al [14] was used to describe Model I:

$$PSA_t = BAS * e^{G*t_s}(e^{-D*t_k} + e^{G*t_k} - 1) \tag{1}$$

where $PSA_t$ is the tumor burden represented by the PSA level at time $t$ (in days); $BAS$ is the baseline PSA; $D$ is the drug effect on PSA, which decays over time due to the development of drug resistance; and $G$ is the rate of growth of PSA due to proliferating tumor. $T_s$ represents the time of PSA growth before the initiation of LHRH treatment; $t_s = t$ if $t \leq$ time of the first LHRH dose ($t_1$) and $t_s = t_1$ if $t > t_1$. $T_k$ represents the time after the initiation of LHRH treatment; $t_k = 0$ if $t_k \leq t_1$ and $t_k = t-t_1$ if $t > t_1$.

It is widely recognized that tumor is highly heterogeneous, thus drug resistance can occur from treatment-induced selection of a subpopulation of drug-resistant cancer cells that existed in the original tumor [34, 35]. Therefore, Model II in Eqs 2 and 3 was used to describe this type of drug resistance:

$$PSA_{Rt} = BAS * (R) * e^{G_R*t} \tag{2}$$

$$PSA_{St} = BAS * (1 - R) * e^{G_S*t_s} * e^{-D_S*t_k} \tag{3}$$

where $PSA_{RT}$ and $PSA_{ST}$ represent PSA levels at time $t$ produced by drug-resistant and drug-sensitive cancer cell population, respectively. $R$ is the proportion of drug-resistant cancer cells in the initial tumor population. $R$ is expressed as $e^{-RP}$ in the modeling process in order to obtain values of $R$ between 0 and 1. $G_R$ and $G_S$ are growth rates of PSA due to replicating drug-resistant and drug-sensitive cancer cells, respectively. $D_S$ is the drug effect on PSA levels due to killing of the drug-sensitive cancer cells. $T_s$ represents the time of PSA growth before the initiation of LHRH treatment; $t_s = t$ if $t \leq t_1$ and $t_s = t_1$ if $t > t_1$. $T_k$ represents the time after the initiation of LHRH treatment; $t_k = 0$ if $t_k \leq t_1$ and $t_k = t-t_1$ if $t > t_1$.

Inter-individual variability in the studied population was modeled for all parameters as follows:

$$\theta_i = \theta_{Typical} e^{\eta_i} \tag{4}$$

where $\theta_i$ is a model parameter, $\theta_{Typical}$ is the typical value of the corresponding parameter in the population, and $\eta_i$ is a normally distributed random effect with a mean of 0 and a variance of $\omega_i^2$. The residual error model in the analysis was additive error as follows:

$$DV_{ij} = \widehat{DV_{ij}} + \epsilon_{ij} \tag{5}$$

where $DV_{ij}$ and $\widehat{DV_{ij}}$ stand for $j_{th}$ log observed and predicted concentration for the $i_{th}$ subject. $\varepsilon_{ij}$ follows normal distribution with a mean of 0 and a variance of $\sigma_{add}^2$. Proportional error model and mixed error model were also assessed in the analysis.

Data were analyzed by nonlinear mixed effects modeling utilizing the Monte Carlo Parametric Expectation Maximization (MCPEM) method in NONMEM software (version 7.3, ICON Development). The details of the MCPEM algorithm have been presented elsewhere [36–38]. Briefly, two-stage hierarchical nonlinear mixed effects modeling was used to find the optimal population mean μ and variance Ω that best describe the observed data. Final population parameters μ and Ω were obtained by first evaluating the conditional mean $\bar{\theta}_i$ and

conditional variance $\bar{B}_i$ for each subject using fixed values of μ and Ω (the expectation step E) according to Eqs 6 and 7, followed by evaluating updates to μ and Ω using Eqs 8 and 9 (the maximization step M) [37–39].

Expectation (E) step:

$$\bar{\theta}_i = \frac{\sum_{k=1}^{N} \theta_k W(l_i(\theta_k), h(\theta_k))}{\sum_{k=1}^{N} W(l_i(\theta_k), h(\theta_k))} \tag{6}$$

$$\bar{B}_i = \frac{\sum_{i=1}^{N} (\theta_k - \bar{\theta}_i)(\theta_k - \bar{\theta}_i)' W(l_i(\theta_k), h(\theta_k))}{\sum_{i=1}^{N} W(l_i(\theta_k), h(\theta_k))} \tag{7}$$

Maximization (M) step:

$$\mu = \frac{1}{m} \sum_{i=1}^{m} \bar{\theta}_i \tag{8}$$

$$\mu = \frac{1}{m} \sum_{i=1}^{m} (\bar{\theta}_i - \mu)(\bar{\theta}_i - \mu)' + \frac{1}{m} \sum_{i=1}^{m} \bar{B}_i \tag{9}$$

where $l_i(\theta_k)$ is the likelihood function for subject $i$ regarding to parameter $\theta_k$ given data; $h(\theta_k)$ is the density function of $\theta_k$ given μ and Ω. The weight $W$ depends on likelihood function, $l_i(\theta_k)$, density function (suppressing dependence on μ and Ω), and the method of Monte-Carlo method used, for $N$ randomly generated parameter vectors of $\theta_k$. $m$ represents the total number of subjects in the analysis. E and M steps were repeated until μ and Ω no longer change [36, 37]. At this point, final population parameters μ and Ω that best described the data were obtained. In MCPEM, the Monte-Carlo integration method was used to evaluate $\bar{\theta}_i$ and $\bar{B}_i$ during the expectation step [36, 37].

The parameters of the developed models were obtained by fitting the models to PSA levels from all subjects simultaneously. Comparison of alternative nested structural models was based on the typical goodness-of-fit diagnostic plots and likelihood ratio test [40, 41]. When comparing alternative nested hierarchical models, the differences in objective function value (OFV), defined as *-2log(likelihood)*, are approximately chi-square distributed with $n$ degrees of freedom ($n$ is the difference in the number of parameters between the full and the reduced model) [41]. Differences in OFV of greater than 10.83 for one degree of freedom, corresponding to a significance level of 0.001, were used to discriminate two nested hierarchical models. This stringent criterion was used because of multiple comparisons inherent in the model selection procedure and random noise associated with the Monte-Carlo sampling technique employed in the MCPEM algorithm [39, 40, 42]. The number of simulated Monte-Carlo parameter sets (ISAMPLE) used for the E step evaluation determined the random noise associated with model parameters and likelihood estimation in the MCPEM method. Larger ISAMPLE was associated with lower random noise of likelihood for more reliable model selection but at the expense of increased computation times. Therefore, to achieve the desirable balance between random noise of model parameters/likelihood and computation time, a sequential approach with different ISAMPLE in different stages of the MCPEM was used in the analysis. First, ISAMPLE of 1000 for first 150 EM iterations and then ISAMPLE of 2000 for another 50 EM iterations were used in the burn-in phase of MCPEM to achieve the stationary phase rapidly. Then, ISAMPLE of 50,000 for 50 EM iterations was used to obtain stable and desirable final model parameter estimates using MCPEM model convergence criteria described previously [42]. In brief, last five EM iterations of population mean parameters and inter-subject

variances with ISAMPLE of 50,000 at stationary state were examined using linear regression analysis. A Bonferroni method using the following equation was used to adjust the $p$ value of linear regression analysis for multiple hypothesis testing:

$$\alpha_B = \frac{\alpha}{B} \qquad (10)$$

where $\alpha_B$, $\alpha$, and $B$ were the Bonferroni-adjusted $p$ value, preset $p$ value (set at 0.05), and the number of tested parameters, respectively. If changes in all tested parameters across iterations were not statistically different from zero, then model convergence is assumed. To compare non-nested models such as Models I and II in this study, the following Akaike information criterion (AIC) was used to select the best model with lowest AIC value:

$$AIC = OFV + 2 * N_{PAR} \qquad (11)$$

where $N_{PAR}$ is the total number of model parameters.

PSA concentrations below 0.1 ng/ml (lower limit of assay quantification) were treated as fixed point censored observations, and the maximum likelihood was used to fit the model to the censored observations [42, 43]. In this case, the likelihood for all data is maximized with respect to model parameters, and the likelihood for a censored concentration was taken to be the likelihood that the censored observation is truly below the limit of quantification. This approach allowed PSA levels <0.1 ng/mL to be included into model development to better characterize the PSA kinetics and treatment effect of leuprorelin.

## Covariate analysis

In order to investigate and quantify relationships between important model parameters and patient-specific factors (covariates), final covariate model was developed using a modified Wald's approximation method with backward elimination (WAM-BE) proposed by our group [44, 45]. In brief, all potential covariate-parameter relationships were incorporated into the best structural model to form the full model with covariates. Parameter estimates and the covariance matrix (COV) from the full model fit were used to calculate the Wald's approximation statistics. Assuming the vector of fixed-effect covariate parameters was $k \times 1$ vector $\boldsymbol{\theta}$, and the corresponding COV for these parameters was $k \times k$ matrix C. $\boldsymbol{\theta}$ could be partitioned to $p \times 1$ vector $\boldsymbol{\theta_1}$ and $q \times 1$ vector $\boldsymbol{\theta_2}$, where $\boldsymbol{\theta_1}$ were fixed-effect covariate parameters that need to be estimated and $\boldsymbol{\theta_2}$ were covariate parameters restricted to zeros under the hypothesized submodel. Then $\boldsymbol{\theta}$ and the corresponding C could be defined as follows:

$$\boldsymbol{\theta} = \begin{pmatrix} \boldsymbol{\theta}_1 \\ \boldsymbol{\theta}_2 \end{pmatrix} \text{ and } C = \begin{pmatrix} C_{11} & C_{12} \\ C_{21} & C_{22} \end{pmatrix} \qquad (12)$$

The Wald's approximation to the likelihood ratio test (LRT) statistic ($\Lambda'$) for the hypothesized submodel that $\boldsymbol{\theta}_2 = 0$ was described by the equation:

$$\Lambda' = \boldsymbol{\theta}_2' C_{22}^{-1} \boldsymbol{\theta}_2 \qquad (13)$$

It has been shown that maximum likelihood estimates of $\boldsymbol{\theta}$ followed asymptotically multivariate normal distribution with covariance matrix C [46]. Under $H_0$: $\boldsymbol{\theta}_2 = 0$, the asymptotical distribution of $\Lambda'$ followed $\chi_q^2$ distribution, and hypothesis testing could be applied to decide whether or not to reject $H_0$. The calculation of $\Lambda'$ in Eq 13 did not require running the submodels, and therefore, multiple hypotheses could be efficiently tested by using results from the full model fit. In WAM-BE method, the backward elimination (BE) process was used to screen

and eliminate insignificant covariates from the full model based on the difference in values of $\Lambda'$ between two models. Differences in $\Lambda'$ of greater than 10.83 for 1 degree of freedom, corresponding to a significance level of 0.001, were used to discriminate two nested hierarchical models. The best models selected by BE approach based on $\Lambda'$ values were then used as the starting models for selecting the final covariate model using actual NONMEM runs with the BE process.

Continuous covariates were added into the structural model as follows [40]:

$$\theta_i = \theta_{Typical}\left(\frac{COV_j}{median(COV_j)}\right)^{\theta_{covij}} \tag{14}$$

where $COV_j$ is the value of $j_{th}$ continuous covariate, and $\theta_{covij}$ is the effect the $j_{th}$ covariate has on the $i_{th}$ parameter.

Categorical covariates were added as follows:

$$\theta_i = \theta_{Typical} * e^{(COV_j*\theta_{covij})} \tag{15}$$

where $COV_j$ is an indicator variable (e.g., if the categorical variable is gender, $COV_j = 1$ when the subject is male, and $COV_j = 0$ when the subject is female).

The final model was evaluated by routine diagnostic plots. Additionally, prediction-corrected visual predicted check (VPC) was done in PsN (ver. 4.9.0, Uppsala University, Sweden) using 1000 simulations and automatic binning, to assess the performance of the final model [47]. The final population model was used to examine the effects of statistically significant covariates on clinical outcomes associated with leuprorelin treatment. PSA progression is defined by the Prostate Cancer Clinical Trials Working Group (PCWG2 and PCWG3) as "the date that a 25% or greater increase and an absolute increase of 2 ng/mL or more from the nadir is documented, which is confirmed by a second value obtained 3 or more weeks later" [48, 49]. The percentages of leuprorelin-treated subjects with PSA progression within one, two, and three years simulated from 1000 subjects using 5th percentile, median and 95th percentile values of continuous covariates and different categories of categorical variables were obtained and compared. Simulation was conducted with the final population model using R.

The parallel computing platform of NONMEM was implemented in a single workstation equipped with dual Intel Xeon E5-2698 v4 20-cores CPU with 2.20 GHz, a Windows 10 Enterprise operating system and Intel Parallel Studio XE 2016 Fortran compiler.

## Results

A total of 1113 PSA observations from 264 subjects were used for model development. Baseline demographic data and laboratory values of all subjects are summarized in Table 1. The study population had a median age of 80 and median baseline PSA of 8.5 ng/mL. There was no missing baseline demographic data and laboratory values among subjects included in the analysis.

Model I that assumed drug resistance was developed from PSA-producing PCa cells that were initially sensitive to LHRH treatment, had an AIC value of 865.432. Model II that assumed drug resistance was originated from treatment-induced selection of a subpopulation of drug-resistant PSA-producing cancer cells, had a lower AIC of 295.671. Therefore, Model II was selected as the structural model. Use of proportional or mixed error models resulted in unstable models, so an additive error model was used. HGB was a significant covariate that affects the proportion of drug-resistant cells ($R$ or $e^{-RP}$) in the original tumor. Baseline PSA (BAS) and antiandrogen use (AND) were significant covariates on the drug killing effect of drug-sensitive PSA-producing cancer cell population ($D_S$). Addition of these three covariates

**Table 1. Summary of baseline demographic data and laboratory values of subjects included in model development.**

| Baseline characteristics | Median (range) or counts |
|---|---|
| Age (years) | 80 (60–100) |
| Race | |
| Caucasian | 189 |
| Black | 59 |
| Hispanic/other | 16 |
| Region | |
| South | 196 |
| West | 20 |
| Midwest | 42 |
| Northeast | 6 |
| Antiandrogen use | |
| Yes | 33 |
| No | 231 |
| [a]AST (IU/L) | 20 (9–91) |
| [b]ALT (IU/L) | 18 (4–110) |
| [c]SCR (mg/dL) | 1.10 (0.700–9.30) |
| [d]ALP (IU/L) | 76.5 (23.0–3640) |
| [e]ALB (g/dL) | 4.13 (2.90–4.80) |
| [f]HGB (g/dL) | 13.6 (6.80–17.4) |
| Baseline [g]PSA (ng/mL) | 8.50 (0.200–782) |

Data are presented as median (range) or counts.
[a]aspartate transaminase
[b]alanine transaminase
[c]serum creatinine
[d]alkaline phosphatase
[e]albumin
[f]hemoglobin
[g]prostate-specific antigen.

into the model resulted in significant decrease in OFV (50.008; p<0.001 for degrees of freedom = 3). The final model with covariates was presented as follows:

$$RP = \theta_{RP}\left(\frac{HGB}{13.6}\right)^{\theta_{HGB\_RP}} \qquad (16)$$

$$D_S = \theta_{DS} * \left(\frac{BAS}{8.5}\right)^{\theta_{BAS\_DS}} * e^{IND_{AND}*\theta_{AND\_DS}} \qquad (17)$$

where $\theta_{HGB\_RP}$ and $\theta_{BAS\_DS}$ are model parameters used to describe the effect of HGB on $RP$ and BAS on $D_S$, respectively. $\theta_{AND\_DS}$ is used to describe the effect of AND on $D_S$. $IND_{AND} = 1$ if the patient used antiandrogen within 30 days of leuprorelin initiation and = 0 otherwise. The parameter estimates of the final model with covariates are presented in Table 2. All model parameters were estimated with good precision (percent coefficient of variation, %CV<50). The population estimate of $D_S$ was 3.78 x 10$^{-2}$ day$^{-1}$. Population estimate of growth rate for drug-sensitive PSA-producing tumor cell population ($G_S$) was 1.96 x 10$^{-3}$ day$^{-1}$, corresponding to a PSA doubling time of 354 days. Population estimate of $RP$ was 3.94, indicating that 1.94%

Table 2. Parameter estimates of the final PSA kinetics model.

| Parameter | Estimate | [a]%CV |
|---|---|---|
| Structural Model | | |
| [b]$D_S$ (day$^{-1}$) | 3.78 x 10$^{-2}$ | 6.19 |
| [c]$Gs$ (day$^{-1}$) | 1.96 x 10$^{-3}$ | 22.5 |
| [d]$RP$ | 3.94 | 7.44 |
| [e]$G_R$ (day$^{-1}$) | 6.54 x 10$^{-4}$ | 28.4 |
| Interindividual Variability | | |
| $\omega^2_{DS}$ | 0.453 | 14.8 |
| $\omega^2_{GS}$ | 2.59 | 19.7 |
| $\omega^2_{RP}$ | 0.944 | 16.4 |
| $\omega^2_{DR}$ | 3.76 | 21.1 |
| Covariate Model | | |
| [f]HGB on $RP$ ($\theta_{HGB\_RP}$) | 2.30 | 24.7 |
| [g]BAS on $D_S$ ($\theta_{BAS\_DS}$) | 0.174 | 24.5 |
| [h]AND on $D_S$ ($\theta_{AND\_DS}$) | 0.677 | 30.0 |
| Residual Variability | | |
| Additive error ($\sigma_{add}$) | 2.01x10$^{-1}$ | 3.49 |

[a]%CV = percent coefficient of variation

[b]drug effect on drug-sensitive tumor cells

[c]growth rate of drug-sensitive cells

[d]exp(-$RP$) represents the fraction of drug-resistant tumor cells in the original tumor

[e]growth rate of drug-resistant cells

[f]effect of hemoglobin level on $RP$

[g]effect of baseline prostate-specific antigen level on $Ds$

[h]effect of antiandrogen use on $Ds$

(calculated from $R = e^{-RP}$) of the original PSA-producing cancer cell population was inherently resistant to leuprorelin treatment. Population estimate of growth rate for drug-resistant PSA-producing cancer cell population ($G_R$) was 6.54 x 10$^{-4}$ day$^{-1}$, corresponding to a PSA doubling time of 1060 days.

Large between-patient variability was observed in these parameters, possibly due to diverse PSA kinetics in the studied population. The estimated value of $\theta_{HGB\_RP}$ was 2.30, suggesting that subjects with lower HGB levels had lower $RP$ and therefore were less likely to respond to leuprorelin treatment. Typical values of $RP$ were 2.37, 3.94, and 5.73 for subjects with 5[th] percentile (10.9 g/dL), median (13.6 g/dL), and 95[th] percentile (16.0 g/dL) levels of HGB, respectively. The corresponding proportions of resistant PSA-producing cancer cell population in the original tumor were 9.36%, 1.94%, and 0.326%, respectively. The estimated value of $\theta_{BAS\_DS}$ was 0.174, indicating that higher baseline PSA levels were associated with better LHRH-treatment effect on PSA level due to killing of drug-sensitive cancer cell population. Typical values of $D_S$ were 2.32 x 10$^{-2}$, 3.78 x 10$^{-2}$, and 5.99 x 10$^{-2}$ day$^{-1}$ for subjects with 5[th] percentile (0.515 ng/mL), median (8.50 ng/mL), and 95[th] percentile (120 ng/mL) levels of BAS, respectively. The estimated value of $\theta_{AND\_DS}$ was 6.77 x 10$^{-1}$, implying that subjects with anti-androgen use within 30 days of leuprorelin initiation was associated with 96.8% higher LHRH treatment effect on killing of drug-sensitive PSA-producing cancer cell population compared to those without antiandrogen use.

As shown in Fig 1, diverse PSA kinetics profiles from different patients were well described by the final model. The diagnostic plots in Fig 2 and the prediction-corrected VPC plot shown

in S2 Fig show that the observed PSA values were generally in agreement with predicted PSA levels, and that the final model provided a good fit for the data. The final model was then used in simulation to assess effects of HGB, BAS and AND on PSA progression within one, two and three years, and the results are shown in Fig 3. The percentages of leuprorelin-treated subjects with PSA progression within one year for subjects with 5th percentile, median and 95th percentile of HGB were 19.8, 13.9 and 10.5, respectively (Fig 3A). 38.8, 28.2, and 22.1% of the leuprorelin-treated subjects with 5th percentile, median and 95th percentile of HGB were expected to experience PSA progression within three years after initiation of leuprorelin treatment. The percentages of leuprorelin-treated subjects with PSA progression within one year in subjects with 5th percentile, median and 95th percentile of BAS were 6.9, 13.9, and 25.6, respectively (Fig 3B). The percentages of leuprorelin-treated subjects with PSA progression within three years in subjects with 5th percentile, median and 95th percentile of BAS were 15.8, 28.2, and 47.8, respectively (Fig 3B). 28.5% of leuprorelin-treated subjects with antiandrogen use and 28.2% of those without antiandrogen use were expected to experience PSA progression within three years of treatment period.

## Discussion

To the best of our knowledge, this was the first study to use medical claims data to construct a population-based disease progression model to describe PSA kinetics in patients with hormone-sensitive PCa treated with leuprorelin. Furthermore, the model could quantify effects of patient-specific factors (covariates) on model parameters and PSA kinetics in these patients. Although leuprorelin was selected as the only form of ADT in this analysis, we would expect that our results could be extrapolated to other LHRH agonists, given the similarity in their efficacy reported in clinical trials [50, 51].

Population data analysis using nonlinear mixed effects modeling approach has been shown to provide precise and robust estimates of drug-related model parameters and their population variability using clinical data [41, 52]. Among different nonlinear mixed effects modeling methods, MCPEM was used to successfully develop the final population disease progression model in this study, due to the ability of this method to precisely approximate the true log-likelihood, and its stability and amenability to efficient parallel computing when dealing with complex models with large datasets [38, 39, 53].

One of the important goals in population-based disease progression modeling is to quantify effects of covariates on model parameters and clinical outcomes. Established quantitative relationships are useful for helping clinicians make conscious decisions on selecting optimal drug treatment, with the ultimate goal of implementing individualized or personalized medicine [54]. However, development of population-based disease progression models with covariate-parameter relationships is both time-consuming and labor-intensive, as many submodels with different covariate-parameter relationships need to be created, tested, and compared. There is a total of $2^{4 \times 11} = 1.76 \times 10^{13}$ possible submodels with different covariate-parameter relationships in this study. Due to the structural model complexity and large number of observations, a single covariate model run with ISAMPLE of 50,000 executed in parallel mode with 25 Intel Xeon 2.2 GHz E5-2968 v4 computing cores took about 10 min to complete. Therefore, it was impossible to explore all the possible submodels in this setting. Hence, the WAM-BE method originated by our group was used to develop covariate models [45]. WAM-BE only requires a single full model fit and does not require fitting the submodels to estimate the difference in OFV between two tested models for covariate selection. In addition, WAM-BE is designed to overcome the inherent limitation of the original WAM method in population data analysis [44] and it achieved comparable results with significantly shorter computational time in

models with large numbers of model parameters and tested covariates. Although the original WAM method uses the Wald approximation to LRT statistic ($\Lambda'$) from the full model instead of actual NONMEM model runs to screen covariate models, it is extremely inefficient in the presence of large numbers of tested model parameters and covariates due to the need to calculate $\Lambda'$ values for all possible covariate models. In this study, 1.76 x $10^{13}$ possible covariate models needed to be screened and tested with the original WAM method. Assuming the time to compute $\Lambda'$ value for a covariate model was about $10^{-6}$ seconds, a total of 1.76 x $10^7$ seconds or 203 days would be needed to calculate $\Lambda'$ values of all possible covariate models. To overcome this limitation, we used the sequential backward elimination (BE) process to efficiently eliminate any insignificant covariate models to generate the best starting models, which were subject to subsequent actual NONMEM runs with the BE process to select the final covariate model. Therefore, the WAM-BE method significantly reduced the number of covariate models needed to be tested using WAM-derived $\Lambda'$ values and actual NONMEM runs, and it only took about 39 min for our workstation to develop the final covariate model.

ADT is the standard of care for patients with hormone-sensitive PCa, but most patients develop castration resistance within 1–3 years [53]. The underlying mechanism is thought to be multifactorial and involve several molecular and genetic alterations, and the "adaptation" and the "clonal selection" models have been proposed to explain this phenomenon [55]. With the "adaptation" model, early PCa cells are assumed to have similar androgen requirement for survival and growth, and castration resistance stems from genetic or epigenetic alterations in the cells. With the "clonal selection" model, it is assumed that early PCa cells have heterogeneous androgen requirement, and after ADT castration-resistant cells gradually outgrow castration-sensitive cells due to the survival advantage of the former [56].

In our model, PSA was used as a marker of tumor burden in the modeling of disease progression, as PSA levels are routinely used for disease monitoring and surveillance purposes in PCa [20–23], and they are readily retrievable from health claims database. PSA level has been shown to be closely related to the tumor volume [57], and it was found to reflect the androgen milieu in patients with localized PCa on ADT [51]. A rising PSA is usually the first sign of tumor regrowth, followed by worsening of disease identified by imaging and development of clinical symptoms [49]. In this study, Model I was the mathematical representation of "adaptation" model that assumed drug resistance was developed from PSA-producing cancer cells initially sensitive to LHRH treatment. On the other hand, Model II was the "clonal selection" model that assumed drug resistance was developed from a subpopulation of drug-resistant PSA-producing cancer cells in the original tumor. Our predefined model selection criteria determined that Model II was superior than Model I at describing the PSA profiles of our study subjects, and this finding was consistent with the observations from some recent studies [55].

Based on recent recommendations from the PCWG, PSA responses to drug treatment that affect tumor cell kill can be categorized into three distinct patterns: 1) significant and sustained PSA decline after drug treatment for "responders", 2) initial decline followed by a slow PSA increase for "partial responders", and 3) a transient decrease followed by a rapid PSA increase observed in "non-responders" [49]. As shown in Fig 1, the final model in this study was able to adequately describe these three distinct PSA responses to leuprorelin treatment. Furthermore, the model was able to provide additional biological insight into the observed PSA patterns. For example, Subject C in Fig 1C showed least response to treatment and most significant increase in PSA within the follow-up period among the representative subjects. The modeling results suggested that this subject had a high percentage of drug-resistant cancer cell population of 28.1% and high growth rate of the resistant tumor cell population of 7.97 x $10^{-3}$ day$^{-1}$, both of which were more than 10 times higher than the population mean values. On the other hand,

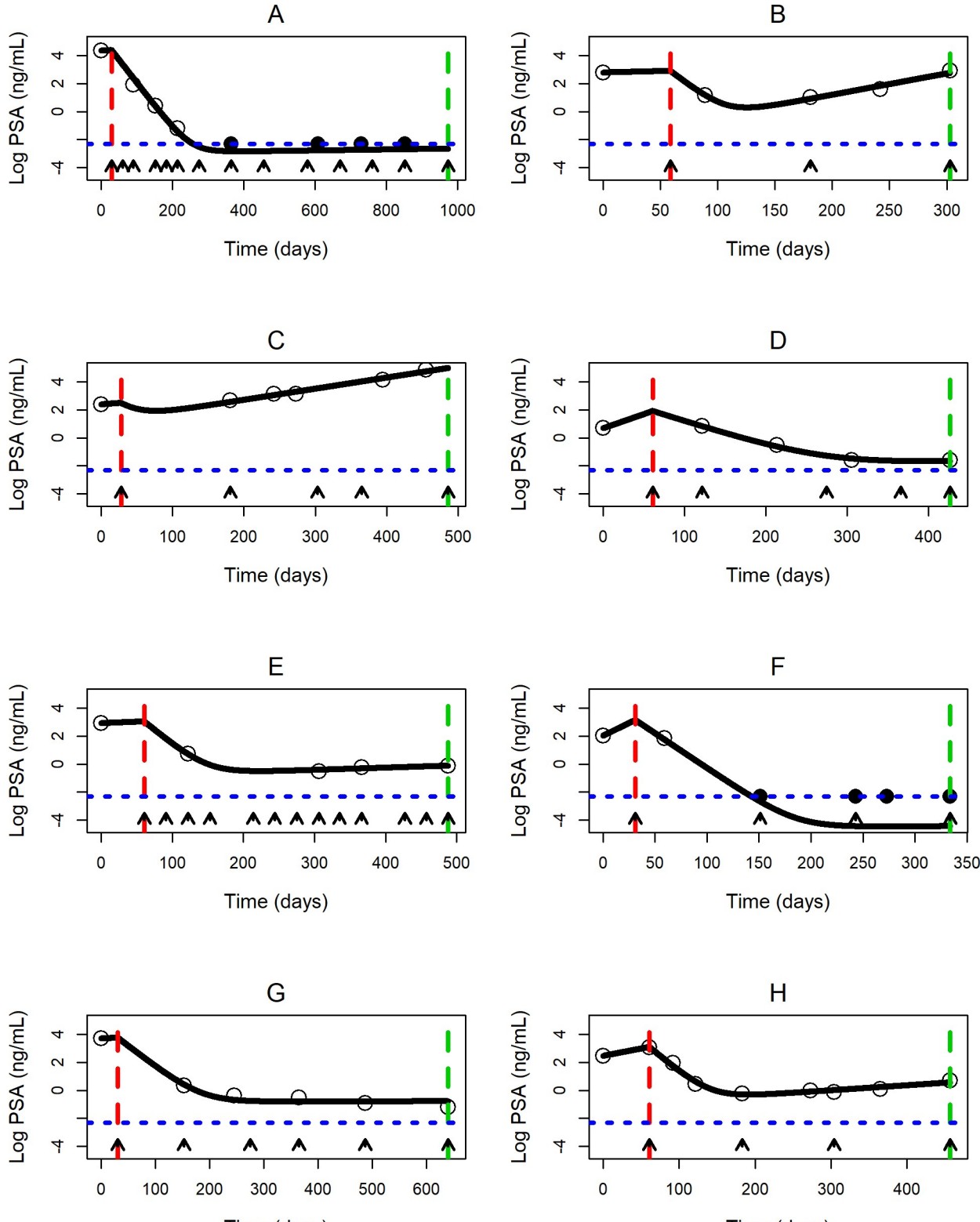

**Fig 1. Profiles of PSA kinetics in representative patients with hormone-sensitive prostate cancer.** Black line represents model-predicted PSA levels. Open circles and solid circles represent observed PSA levels above the lower limit of quantification (LLOQ) and observed PSA levels below the LLOQ, respectively. Red and green vertical dashed lines represent the first and the last recorded date of leuprorelin treatment, respectively. Blue horizontal dotted line represents the LLOQ value of the PSA assay. Black arrows represent the fill dates of leuprolide.

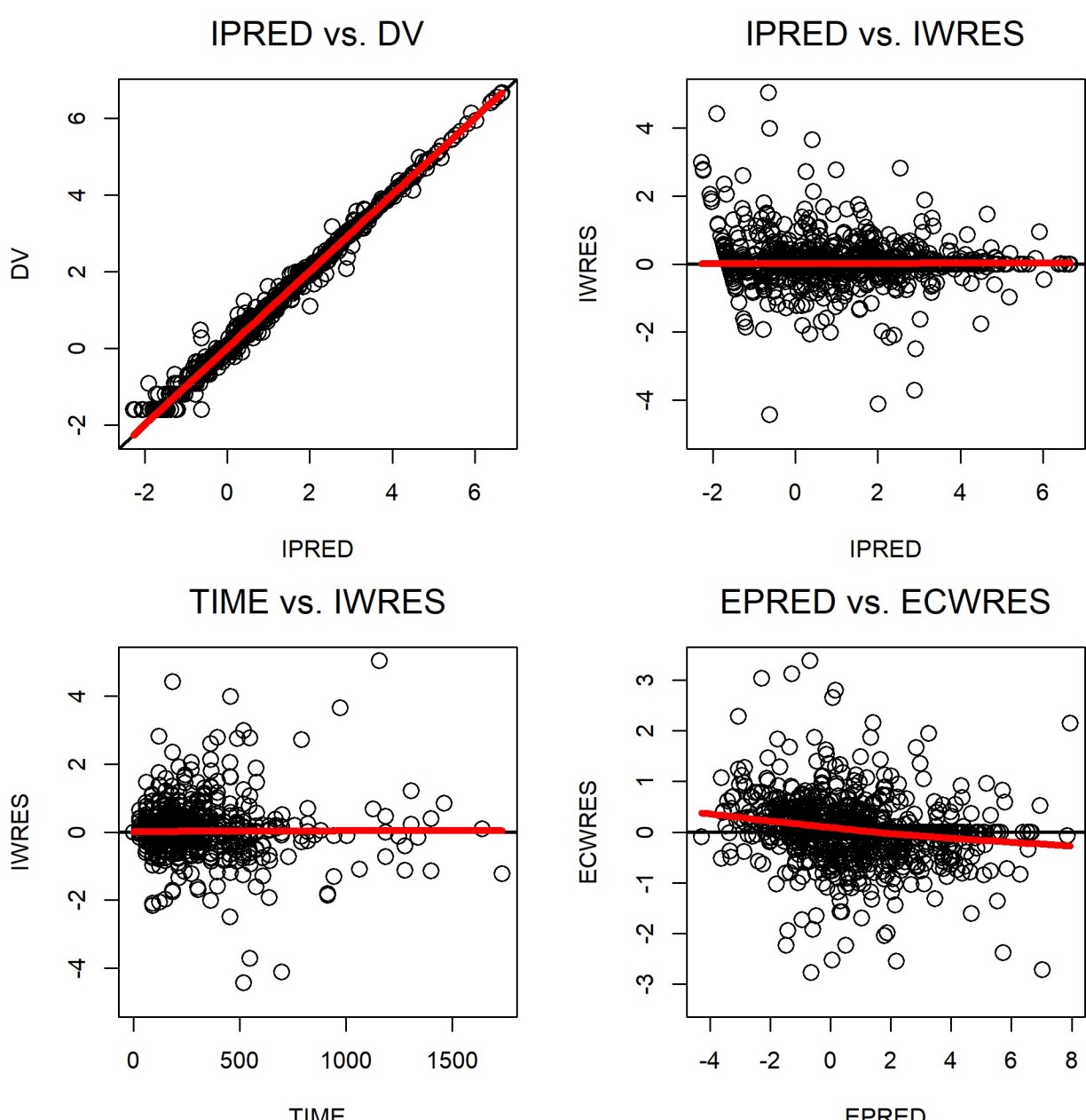

**Fig 2. Diagnostic plots for the final disease progression model.** From left to right and top to bottom: Observed log PSA concentrations (DV) versus individual log predicted values (IPRED), individual weighted residuals (IWRES) versus the IPRED, IWRES versus time, and expected conditional weighted residuals (ECWRES) versus the expected log predicted values (EPRED). The red line represents the loess regression line.

the profile of Subject B in Fig 1B was characterized by a decrease followed by a slow increase of PSA. As shown in Fig 4B, the rapid but transient decline in the total PSA level after leuprorelin treatment was due to the killing of drug-sensitive PSA-producing cancer cell population. However, the high growth rate of drug-resistant PSA-producing cancer cell population of $1.53 \times 10^{-2}$ day$^{-1}$ (compared to the population estimate of $6.54 \times 10^{-4}$ day$^{-1}$) contributed to the subsequent steady rise in PSA level observed in this subject. On the other hand, PSA profile of Subject A in Fig 1A was characterized by significant and sustained PSA decline after leuprorelin treatment and therefore could be described as a "responder". The low percentage of drug-

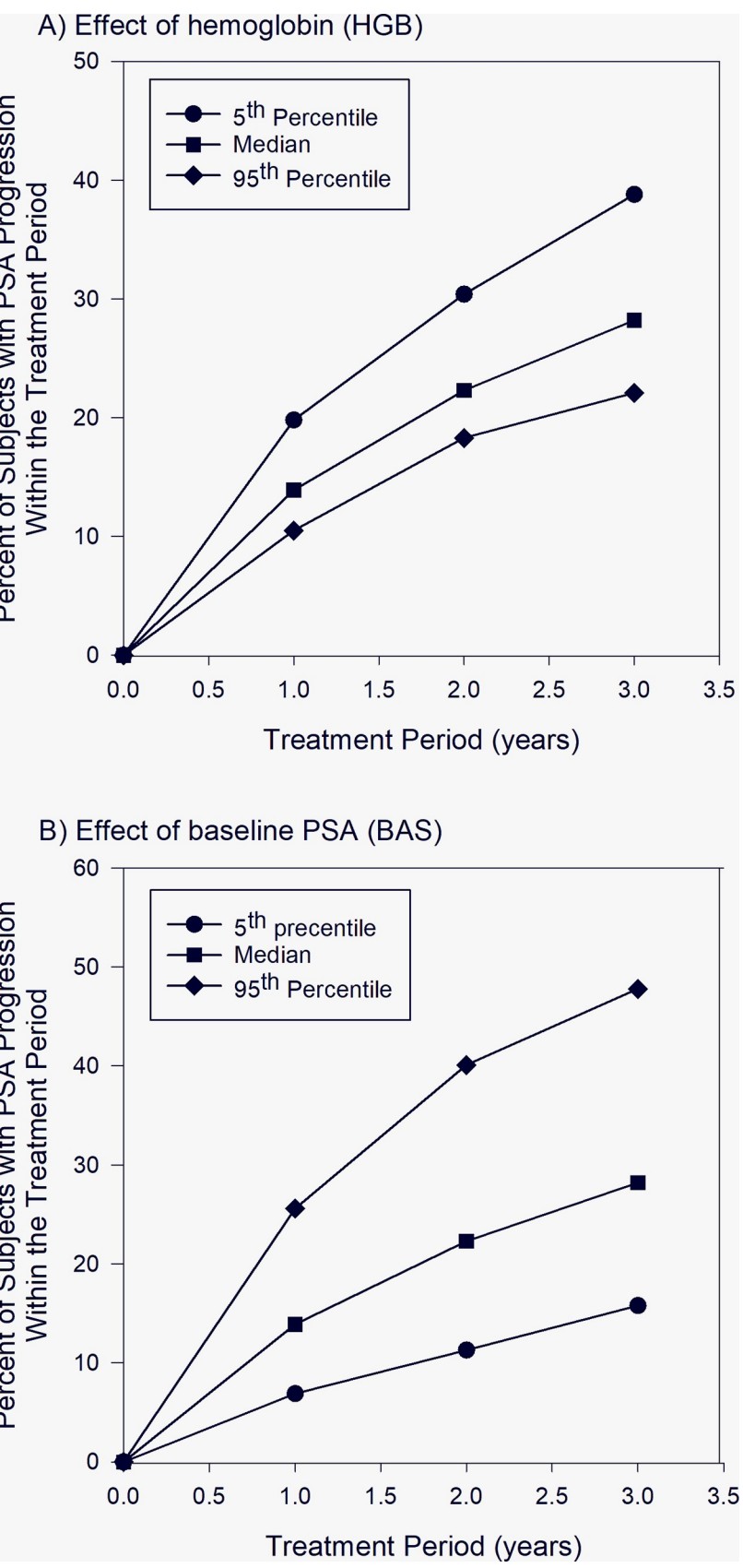

**Fig 3. Simulation results for percentages of subjects with PSA progression within one, two, and three years.** 3A and 3B show simulated PSA progression in subjects with 5th percentile, median and 95th percentile of hemoglobin level and baseline PSA level, respectively.

resistant PSA-producing cancer cell population of $6.36 \times 10^{-2}$% combined with the slow growth rate of the drug-resistant PSA-producing cancer cell population of $3.53 \times 10^{-4}$ day$^{-1}$ contributed to the sustained PSA decline in this subject (Fig 4A).

Only a few studies have attempted to develop predictor models for development of castration resistance in patients with hormone-sensitive PCa. Most of these predictive models were developed using multivariate analysis that provides very little biological insight on the relationship between identified predictors and disease progression [8–12]. Several PSA kinetic parameters such as time to PSA nadir and PSA nadir level have been shown to be significant predictors of disease progression for PCa patients receiving ADT [9–11]. However, the complex quantitative relationship between clinical predictors and PSA kinetics and disease progression in ADT-treated PCa patients remains largely unknown. In this study, we developed the first mechanistic population-based disease progression model that allowed us to investigate such complex relationship. Baseline PSA and HGB were identified as important covariates on PSA kinetics, consistent with previous findings that these two covariates played an important role in the development of castration resistance [8, 9, 12]. However, these two covariates affected different model parameters and hence had very different results on PSA kinetics and biochemical progression. Both baseline PSA and antiandrogen use were statistically significant covariates on drug-killing effect on leuprorelin-sensitive PSA-producing cancer cells, and HGB significantly affected the fraction of cancer cells resistant to leuprorelin treatment. Drug-killing rates on leuprorelin-sensitive PSA-producing cancer cells were higher in subjects with higher baseline PSA levels and those with antiandrogen use within the 30 days of leuprorelin treatment initiation, compared to those with lower baseline PSA levels and those without antiandrogen use. Subjects with lower hemoglobin levels had higher fraction of leuprorelin-resistant cancer cells in the original tumor.

Subsequently, simulation was performed to evaluate the relationship between these covariates and PSA kinetics and disease progression. The results showed that subjects with higher baseline PSA levels were more likely to experience PSA progression (Fig 3B), which may seem counterintuitive given that higher baseline PSA was associated with higher drug-killing effect on leuprorelin-sensitive PSA-producing cancer cells in our model. Further analysis of the simulated results showed that the baseline PSA level had a complicated effect on the overall PSA kinetics. The simulated median time to nadir levels was 294, 198, and 135 days after initiation of leuprorelin in subjects with 5th percentile (0.515 ng/mL), median (8.50 ng/mL) and 95th percentile (120 ng/mL) of baseline PSA levels, respectively. The simulated median nadir levels were 0.0220, 0.290, and 3.47 ng/mL for leuprorelin-treated subjects with 5th percentile, median and 95th percentile of baseline PSA levels, respectively. Therefore, subjects with higher baseline PSA levels had higher velocity of PSA decline and shorter time to achieve the nadir level after treatment initiation, which was consistent with higher drug-killing effect on leuprorelin-sensitive cancer cells in these subjects. Furthermore, subjects with higher baseline PSA levels also had higher nadir PSA levels, possibly due to presence of higher numbers of leuprorelin-resistant cancer cells, and therefore they were more likely to experience PSA progression. Overall, our results confirmed findings from previous studies that associated higher baseline PSA levels with shorter time to castration resistance, and they were also in agreement with observations from Ji et al that higher PSA nadir, higher velocity of PSA decline and shorter time to PSA nadir were predictive of increased risk of progression to castration resistance [11]. On the

other hand, subjects with lower levels of HGB were shown to have a higher chance of PSA progression (Fig 3A), consistent with results from previous studies [8, 12]. Mechanistically, the simulated median time to nadir levels were 155, 198, and 240 days after initiation of leuprorelin in subjects with 5th percentile (10.9 g/dL), median (13.6 g/dL) and 95th percentile (16.0 g/dL) of HGB levels, respectively. The simulated median nadir levels were 1.19, 0.290, and 0.0574 ng/mL for subjects with 5th percentile, median and 95th percentile of HGB levels, respectively. Again, we could see that subjects with lower HGB levels had higher PSA nadir, higher velocity of PSA decline and shorter time to PSA nadir, which were the identified PSA kinetic risk factors predictive of shorter time to biochemical progression. Lastly, while use of antiandrogen within 30 days of initiation of leuprorelin treatment was identified as a statistically significant covariate affecting drug-killing effect on leuprorelin-sensitive PSA producing tumor cells, it had minimum effect on PSA progression in our simulation. One plausible explanation was that as antiandrogen use could only be temporary in the study population, the observed inhibitory effect was most likely on PSA flare alone, which has not been found to be associated with tumor progression and negative outcomes [58].

There are several limitations to our study. First, while medical claims database can be used for longitudinal population studies to address questions in real-life clinical practice, the highly heterogeneous nature of the database posed technical challenges to data analysis. For example, patients on both continuous and intermittent regimens of leuprorelin were included in the analysis, as it was difficult to quantitatively identify and separate these two types of treatment regimens based on claims data alone. Additionally, because of the heterogeneity of dosing regimens in the studied population, patient compliance could not be accurately assessed. Intermittent ADT has been extensively tested in PCa patients since the 1980s, mainly due to evidence of its anticancer efficacy and its ability to reduce ADT-related adverse effects [59]. With intermittent ADT, the patient receives ADT for several months consecutively until PSA levels drop below a predetermined threshold, after which the patient enters an off-treatment period [60]. Patients receiving the two treatment modalities may have different patterns of disease progression. Nevertheless, two randomized trials showed that intermittent ADT was noninferior to continuous ADT in terms of survival endpoints in PCa patients [61, 62], while the largest trial so far comparing continuous and intermittent ADT could not conclude or exclude noninferiority of intermittent ADT [63]. In our preliminary analysis, dose intensity (leuprolide dose received normalized to an expected dose of 7.5 mg per month) was calculated. As expected, dose intensity varied greatly among patients on continuous and intermittent ADT, and it did not have statistically significant effect on model parameters. Additionally, exclusion of a large proportion of patients (86.4%) diagnosed with malignant PCa and treated by leuprorelin in our effort to build an explainable model, as well as the presence of large between-patient variability observed in some model parameter estimates, may pose a limitation to interpretation of our findings. Nevertheless, retrospective nature of this study suggested that our findings will need to be validated in prospective clinical studies. Second, some clinical measures that assess disease severity and/or metastasis spread, including metastasis stage, the Gleason score, Eastern Cooperative Oncology Group (ECOG) performance status and the Soloway score, were shown to be predictive of biomedical progression [9, 11, 12]. However, they were not available in the medical claims database, and therefore they could not be evaluated for their importance in our model. Similarly, though LHRH agonists have been assumed to cause testosterone suppression [64], testosterone levels were not available in our patient database for us to investigate the possibility of incomplete testosterone suppression and its potential relationship with PSA kinetics. Third, as survival data were not available in our database, analyses linking PSA kinetics and biomedical progression to survival endpoints were not conducted in this analysis. Last but not least, though PSA level has been previously shown to be significantly associated with

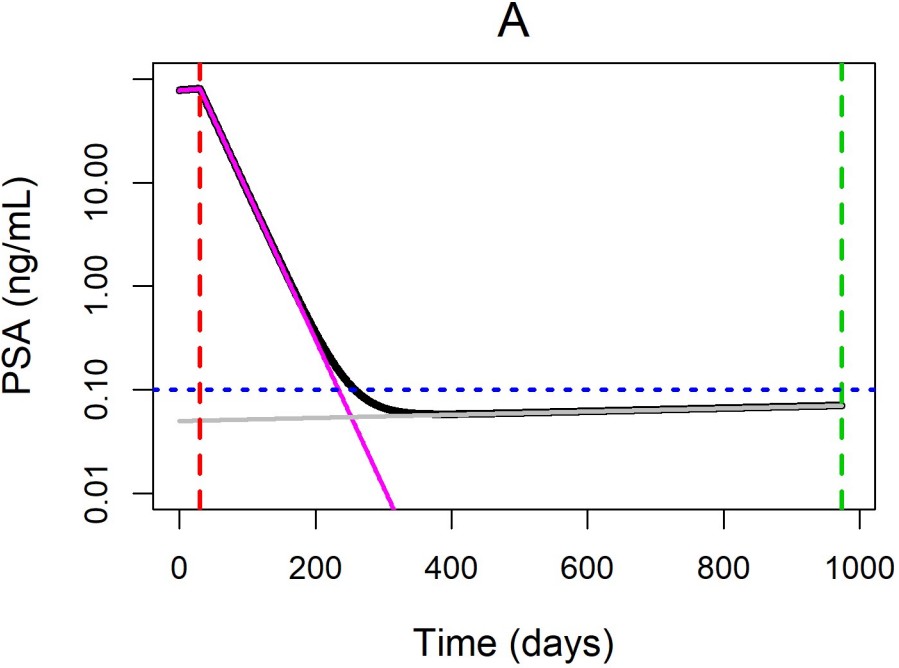

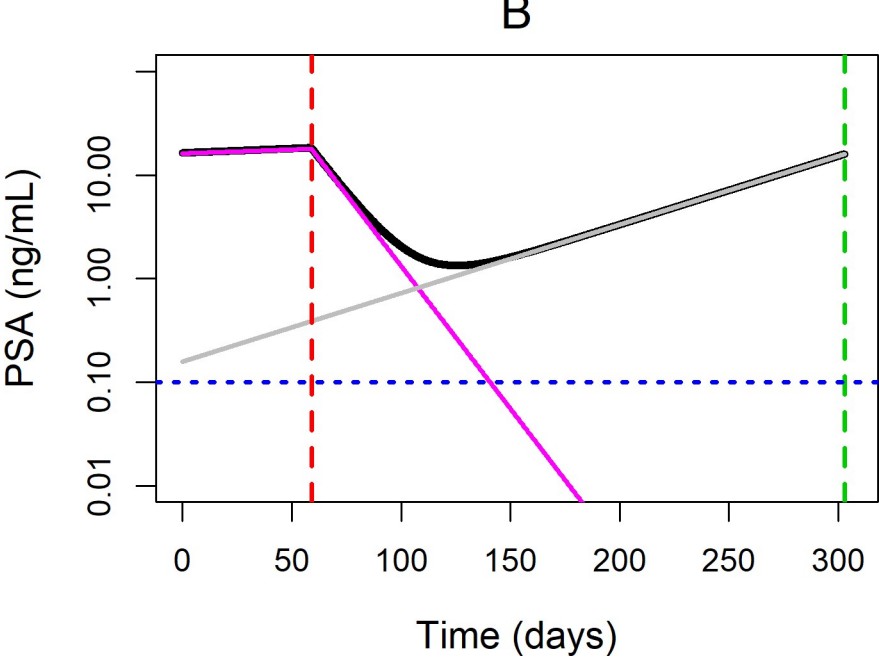

**Fig 4. Simulated total PSA and PSA produced by drug-sensitive and drug-resistant tumor cells in representative subjects.** 4A and 4B demonstrate simulated PSA levels in a "responder" Subject A and a "partial responder" Subject B, respectively. Black line represents model predicted total PSA level. Pink and grey solid lines represent model predicted PSA from drug-sensitive and drug-resistant tumor cells, respectively. Red and green vertical dashed lines represent the first and the last recorded date of leuprorelin treatment, respectively. Blue horizontal dotted line represents the lower limit of quantification (LLOQ) of the PSA assay.

tumor volume [57], it is not as accurate as quantitative imaging biomarkers in representing the tumor burden, and the PSA response does not always associate with treatment response or survival [65].

## Conclusions

In summary, in this study a novel population-based modeling approach was used to provide the first mechanistic insight on how PSA kinetics contributed to the subsequent development of castration resistance in patients with hormone-sensitive PCa treated with leuprorelin. Compared to previous studies, our study successfully modeled the underlying resistance mechanism of PCa cells and provided new biological insight into PSA kinetics in this patient population of interest. The application of population-based disease progression model to existing clinical data allowed estimation of tumor resistant patterns and growth/regression rates that could greatly enhance our understanding of how hormone-sensitive PCa responds to LHRH agonists. It may serve as a platform for incorporating more comprehensive health data in the future, including laboratory measurements, genomic and proteomic data to further personalized medicine in patients with hormone-sensitive PCa.

## Supporting information

**S1 Fig. Patient selection flow diagram.**
(TIF)

**S2 Fig. Prediction-corrected visual predictive check of the final model.** Blue circles represent prediction-corrected observations (log-transformed PSA concentrations). The solid red line represents the median prediction-corrected observations, and the semitransparent red area represents the simulated 95% confidence interval for the median. The dashed red lines represent the 5% and 95% percentiles of prediction-corrected observations, and the semitransparent blue areas represent their respective simulated 95% confidence intervals.
(TIF)

**S1 File. Data dictionary for the Humana dataset.**
(XLSX)

## Author Contributions

**Conceptualization:** Fei Tang, Chee M. Ng.

**Data curation:** Yixuan Zou.

**Formal analysis:** Yixuan Zou, Chee M. Ng.

**Methodology:** Yixuan Zou, Chee M. Ng.

**Resources:** Jeffery C. Talbert.

**Supervision:** Chee M. Ng.

**Visualization:** Yixuan Zou, Chee M. Ng.

**Writing – original draft:** Yixuan Zou, Fei Tang, Chee M. Ng.

**Writing – review & editing:** Fei Tang, Chee M. Ng.

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
