## [Decision Letter · Decision Letter 0]

4 Dec 2019

PONE-D-19-27455

Using medical claims database to develop a population disease progression model for leuprorelin-treated subjects with hormone-sensitive prostate cancer

PLOS ONE

Dear Dr. Ng,

Thank you for submitting your manuscript to PLOS ONE. After careful consideration, we feel that it has merit but does not fully meet PLOS ONE’s publication criteria as it currently stands. Therefore, we invite you to submit a revised version of the manuscript that addresses the points raised during the review process.

Please follow the suggestions of the two reviewers who are experts in prostate cancer field and revise the manuscript to answer their questions.

We would appreciate receiving your revised manuscript by Jan 18 2020 11:59PM. To enhance the reproducibility of your results, we recommend that if applicable you deposit your laboratory protocols in protocols.io, where a protocol can be assigned its own identifier (DOI) such that it can be cited independently in the future. For instructions see: http://journals.plos.org/plosone/s/submission-guidelines#loc-laboratory-protocols

We look forward to receiving your revised manuscript.

Kind regards,

Chih-Pin Chuu, Ph.D.

Academic Editor

PLOS ONE

Journal Requirements:

3. In the ethics statement in the manuscript and in the online submission form, please provide additional information about the patient records used in your retrospective study, including: the date range (month and year) during which patients' medical records were accessed.

4.  Thank you for stating the following in the Financial Disclosure section:

The authors received no specific funding for this work.

We note that one or more of the authors are employed by a commercial company: NewGround Pharmaceutical Consulting LLC

Reviewers' comments:

Reviewer's Responses to Questions

**Comments to the Author**

1. Is the manuscript technically sound, and do the data support the conclusions?

Reviewer #1: Yes

Reviewer #2: Yes

2. Has the statistical analysis been performed appropriately and rigorously? 

Reviewer #1: Yes

Reviewer #2: Yes

3. Have the authors made all data underlying the findings in their manuscript fully available?

Reviewer #1: No

Reviewer #2: Yes

4. Is the manuscript presented in an intelligible fashion and written in standard English?

Reviewer #1: Yes

Reviewer #2: Yes

5. Review Comments to the Author

Reviewer #1: This study well demonstrated pharmacokinetic in leuprorelin treatment of prostate cancer from insurance population data.

The authors are trying to mechanistically explain leuprorelin's effect on prostate cancer cell proliferation by PSA kinetics, which may be quite ambitious

But the article did find itself a role in explaining tumor resistant patterns, which can be of great value in selected patients to early modify prostate cancer treatment

Advantages

1. Proposed a good model of drug resistance development by demonstrating individual leuprorelin sensitive and resistant cancer cell growth rate using PSA findings.

2. Reconfirmed important covariates in prostate cancer development including baseline PSA and hemoglobin, clarify important factors associated with leuprorelin treatment

3. Specify the portion of leuprorelin-resistant cancer cell population and its growth rate can improve patient’s follow-up and the early detection of most at risk patients.

4. The data source is unique and stands for the real-world population. No pharmacokinetics of leuprorelin researches with Humana database to date yet.

5. Provide population mean prostate growth rate and resistance portion for future reference.

Shortcomings

1. Large between-patient variability may indicate that few models may not best estimate leuprorelin on drug resistance of prostate cancer,

2. Need to exclude majority of patients (2035/2354 =86.4%) under leuprorelin treatment in order to build up a explanable model, indicated that the model may not be universally applicable.

3. Figure legends can be refined for better understanding eg. Fig.3A legends should add HGB and Fig.3B BAS on the figure.

4. This research did not take into account the incomplete testosterone suppression and its effect on PSA kinetics.

Reviewer #2: This manuscript is a study which looks at disease progression model for patients with hormone sensitive prostate cancer on leuprorelin. The conclusion is that the proportion of the original prostate cancer cells inherently resistant to treatment was estimated to be 1.94%. This is an interesting study and it can be published after some revision.

Strengths of this study:

1 Large sample size.

2 Well data analyzed.

This paper is easy to follow and well structured. But the characters of the prostate cancer patients should be clarified. The prostate cancer staging, Gleason score of prostate cancer and comorbility data are not available in the paper. This is a major limitation of the study. The authors should tell us the distribution of prostate cancer risk factors in these patients.

6. PLOS authors have the option to publish the peer review history of their article (what does this mean?). If published, this will include your full peer review and any attached files.

Reviewer #1: No

Reviewer #2: No

---

## [Author Response · Author response to Decision Letter 0]

17 Feb 2020

Response to Reviewers

Reviewer #1 Comments

This study well demonstrated pharmacokinetic in leuprorelin treatment of prostate cancer from insurance population data

The authors are trying to mechanistically explain leuprorelin's effect on prostate cancer cell proliferation by PSA kinetics, which may be quite ambitious

But the article did find itself a role in explaining tumor resistant patterns, which can be of great value in selected patients to early modify prostate cancer treatment

We would like to thank the reviewer’s comments, and we have made some changes to our manuscript accordingly.

Advantages

1. Proposed a good model of drug resistance development by demonstrating individual leuprorelin sensitive and resistant cancer cell growth rate using PSA findings.

2. Reconfirmed important covariates in prostate cancer development including baseline PSA and hemoglobin, clarify important factors associated with leuprorelin treatment

3. Specify the portion of leuprorelin-resistant cancer cell population and its growth rate can improve patient’s follow-up and the early detection of most at risk patients.

4. The data source is unique and stands for the real-world population. No pharmacokinetics of leuprorelin researches with Humana database to date yet.

5. Provide population mean prostate growth rate and resistance portion for future reference.

Shortcomings

1. Large between-patient variability may indicate that few models may not best estimate leuprorelin on drug resistance of prostate cancer

We acknowledge that the presence of large between-patient variability is a limitation of our study, and we have included this in our discussion section L621-625.

2. Need to exclude majority of patients (2035/2354 =86.4%) under leuprorelin treatment in order to build up a explanable model, indicated that the model may not be universally applicable.

We acknowledge that the need to exclude the majority of patients is a limitation of our study, and we have included this our discussion section L621-625, which reads, “Additionally, exclusion of a large proportion of patients (86.4%) diagnosed with malignant PCa and treated by leuprorelin in an effort to build an explainable model, as well as the presence of large between-patient variability observed in some model parameter estimates, may pose a limitation to interpretation of our findings.”

3. Figure legends can be refined for better understanding eg. Fig.3A legends should add HGB and Fig.3B BAS on the figure.

Figures 3A and 3B have been corrected to include HGB and BAS on the figures.

4. This research did not take into account the incomplete testosterone suppression and its effect on PSA kinetics.

Testosterone levels are not available in our dataset, as it was not routinely monitored during the period covered by our database (2007-2011). We have acknowledged this limitation in our discussion section L631-634, which reads, “Similarly, though LHRH agonists have been assumed to cause testosterone suppression [63], testosterone levels were not available in our patient database for us to investigate the possibility of incomplete testosterone suppression and its relationship with PSA kinetics.”

Reviewer #2: This manuscript is a study which looks at disease progression model for patients with hormone sensitive prostate cancer on leuprorelin. The conclusion is that the proportion of the original prostate cancer cells inherently resistant to treatment was estimated to be 1.94%. This is an interesting study and it can be published after some revision.

We would like to thank the reviewer’s comments, and please see our response below.

Strengths of this study:

1 Large sample size.

2 Well data analyzed.

This paper is easy to follow and well structured. But the characters of the prostate cancer patients should be clarified. The prostate cancer staging, Gleason score of prostate cancer and comorbility data are not available in the paper. This is a major limitation of the study. The authors should tell us the distribution of prostate cancer risk factors in these patients.

 The prostate cancer staging and Gleason score of the patients are, unfortunately, not available in our dataset, as they were not associated with any diagnostic/procedural/laboratory codes and therefore were not captured by de-identified data extraction from the electronic health records. These limitations were acknowledged in the discussion section L626-631. 

Though comorbidities influence survival in patients with metastatic prostate cancer because they are associated with non-cancer associated deaths, to our knowledge, the effect of comorbidities is on biomedical progression (i.e., progression to castration resistance) has not been reported in the literature. As we focused on biomedical progression in castration-sensitive prostate cancer patients instead of survival in patients with castration-resistant disease, we think that comorbidities are likely not as important than the other factors mentioned and studied in our manuscript, including demographics and laboratory values. Additionally, electronic health record (EHR) problem lists have been shown to have poor sensitivity for detecting major comorbidities compared to free-text notes in the EHR (Daskivich TJ et al, Am J Manag Care. 2018 Jan 1;24(1):e24-e29); however, the latter was not accessible to the authors due to the de-identified nature of the data source.

---

## [Editor Report · Decision Letter 1]

4 Mar 2020

Using medical claims database to develop a population disease progression model for leuprorelin-treated subjects with hormone-sensitive prostate cancer

PONE-D-19-27455R1

Dear Dr. Ng,

We are pleased to inform you that your manuscript has been judged scientifically suitable for publication and will be formally accepted for publication once it complies with all outstanding technical requirements.

With kind regards,

Chih-Pin Chuu, Ph.D.

Academic Editor

PLOS ONE
---

## [Editor Report · Acceptance letter]

9 Mar 2020

PONE-D-19-27455R1 

Using medical claims database to develop a population disease progression model for leuprorelin-treated subjects with hormone-sensitive prostate cancer 

Dear Dr. Ng:

I am pleased to inform you that your manuscript has been deemed suitable for publication in PLOS ONE. Congratulations! Your manuscript is now with our production department. 

With kind regards,

on behalf of

Prof. Chih-Pin Chuu 

Academic Editor

PLOS ONE